# Heterocyclization of Bis(2-chloroprop-2-en-1-yl)sulfide in Hydrazine Hydrate–KOH: Synthesis of Thiophene and Pyrrole Derivatives

**DOI:** 10.3390/molecules27206785

**Published:** 2022-10-11

**Authors:** Igor B. Rozentsveig, Valentina S. Nikonova, Victor V. Manuilov, Igor A. Ushakov, Tatyana N. Borodina, Vladimir I. Smirnov, Nikolay A. Korchevin

**Affiliations:** 1A. E. Favorsky Irkutsk Institute of Chemistry, Siberian Branch of the Russian Academy of Sciences, 1 Favorsky St., 664033 Irkutsk, Russia; 2Chemistry Department, Irkutsk State University, Karl Marx Str., 1, 664003 Irkutsk, Russia

**Keywords:** 2,3-dichloropropene, sulfur, hydrazine hydrate, bis(chloroprop-2-en-1-yl)sulfide, allenic sulfides, acetylenic sulfides, thiiranium ions, migration of thio group, heterocyclization, thiophene derivatives, pyrrole derivatives

## Abstract

The article is devoted to heterocyclization of bis(2-chloroprop-2-en-1-yl)sulfide which proceeds in hydrazine hydrate–alkali medium and leads to formation of thiophene and pyrrole derivatives: previously described 4,5,9,10-tetrahydrocycloocta[1,2-c;5,8-c’]dithiophene, as well as unknown hydrazone of 5-methylidene-3-methyldihydrothiophen-2-one and 1-amino-2-(propynylsulfanylpropenylsulfanyl)-3,5-dimethylpyrrole. Tentative mechanisms for the formation of the heterocyclic products are discussed. Obtained hydrazone of 5-methylidene-3-methyldihydrothiophen-2-one was used for the synthesis of a range of azine derivatives and in oxidation process with SeO_2_. The found reactions open up expedient approaches to the formation of various hardly accessible thiophene and pyrrole compounds from 2,3-dichloropropene and elemental sulfur as starting reagents.

## 1. Introduction

The thiiranium structural motif is an important cationic intermediate that is responsible for chemical transformations in the synthesis of various organosulfur compounds [1,2]. Thiiranium ions are highly reactive electrophiles, which can interact with diverse nucleophiles in a selective fashion that opens new horizons in the preparation of different sulfur-containing linear or heterocyclic derivatives [3,4,5,6,7,8,9,10,11,12,13,14,15]. The well-known anchimeric assistance of chalcogens in case of β-haloalkyl sulfides or –selenides [16] is due to the formation of chalcogeniranium intermediates. Due to anchimeric assistance, the nucleophilic substitution can be increased by thousand folds, and these properties cause specific biological activity (similar to that of “mustard gas-like” systems) [17,18]. Thiiranium intermediates play a crucial role in the mechanism of irreversible inhibition of active sites of coronavirus enzymes inducing severe acute respiratory syndrome (SARS-CoV) [19] that can be employed in design of antiviral drugs.

In addition, chalcogen-containing bridged systems are important objects for theoretical studies using physical-chemical methods and quantum-chemical calculations [20,21,22,23,24].

In the chemistry of organic chalcogenides, the reactions involving 1,2-migration of sulfur via thiiranium and thiirenium systems is of fundamental and applied importance and hold a special place. Such processes are exploited in the synthesis of sulfur-containing carbonyl compounds [15], heterocyclic derivatives or carbocycles, furans and pyrroles [1,4,25,26,27], as well as thiophenes [28] or indenes [29]. Such 1,2-migration of sulfur was discussed in detail by V. Gevorgyan et al. [4,20,21,22,23,24,25]. Summarizing the data given in [1,2,3,4,5,6,7,8,9,10,11,12,13,14,15,16,17,18,19,20,21,22,23,24,25,26,27,28,29] allows the following conclusions to be drawn:(1)Migration of sulfur occurs along either the alkyl fragment (for beta-halo- or beta-hydroxyalkylsulfides), or the cumulene moiety (for sulfanyl-substituted acetylenic or allenic ketones and their imines upon heating in the presence of CuI);(2)In all previous works, the substituent at the sulfur atom is “passively” transferred and not transformed chemically.

Earlier, we [30,31,32,33] developed effective methods for the synthesis of chloropropenyl chalcogenides, precursors of allenes or acetylenic chalcogenides, which are promising building blocks for the preparation of various organochalcogen compounds.

It should be emphasized that 2,3-dichloropropene, the key reagent in the synthesis of organochalcogen compounds by our methodology [30,31,32,33,34,35,36], is a large tonnage waste of organochlorine production, which requires utilization. Another technology-related feedstock is elemental sulfur. Nowadays, the demand for sulfur in market is inferior to the supply. Therefore, the development of new chemistry of 2,3-dichloropropene and elemental sulfur represent an urgent challenge.

In our previous work, we synthesized bis(2-chloroprop-2-en-1-yl)sulfide **1**, which was further cyclized to 4,5,9,10-tetrahydrocycloocta[1,2-c;5,8-c’]dithiophene **2**, as well as other β,β’-substituted thiophene derivatives **3** and **4** [32,33] (Figure 1).

Bis(chloropropenyl)sulfide 1 was obtained from 2,3-dichloropropene and elemental sulfur in the basic-reductive system hydrazine hydrate–KOH, which is successfully employed in the synthesis of organochalcogen compounds [34,35,36]. At the same time, the cyclization of compound 1 to thiophene derivatives effectively proceeded in acetonitrile [32,33].

In continuation of these studies, in the present work, we synthesized new sulfur-containing heterocyclic derivatives from bis(chloropropenyl)sulfide **1** using hydrazine hydrate-KOH as a reaction medium (Figure 1).

## 2. Results

We have found that three heterocyclic compounds are formed from bis(chloropropenyl)sulfide **1** in hydrazine hydrate–KOH medium. Along with the expected dithiophenocyclooctane **2** (in up to 14% yield), hydrazone of 5-methylidene-3-methyl-dihydrothiophen-2-one **5** (in up to 68% yield) and another cyclic product, 1-amino-2-(propynylsulfanylpropenylsulfanyl)-3,5-dimethylpyrrole **6** (in up to 10% yield) are isolated from the reaction mixture (Figure 1).

The yields of products **2**, **5**, and **6** change insignificantly at the reaction temperature (30–60 °C) and KOH: **1** ratio of 3–10. Minimum time at full conversion of the reagent **1** (4 h) was reached at 50 °C. Compounds **2**, **5**, and **6** were isolated by column chromatography and characterized by various physical-chemical methods.

Dithiophenocyclooctane **2** was described in early works [37,38,39,40,41]. In contrast to our previous works [32,33] and results presented herein with bis(chloropropenyl)sulfide **1**, in [37,38,39,40,41], bis(propargyl)sulfide was employed as the key reagent, which was further isomerized to bis(allenyl)sulfide **A**, the latter being isolated individually before involvement in the final reaction. As the authors [37,38,39,40,41] suggested, bis(allenyl)sulfide **A** further undergoes cycloaromatization through the generation of 3,4-dimethylthienyl diradical **B**, which, in turn, dimerizes to afford dithiophenocyclooctane **2** (Figure 2). For bis(chloropropenyl)sulfide **1** as starting reagent ([32,33] and herein), bis(allenyl)sulfide **A** is a possible intermediate produced after dehydrochlorination of **1** (Figure 2).

A tentative mechanism for the formation of dihydrothiophen-2-one hydrazone **5** is shown in Figure 3. Taking into account the structure of the obtained products, it can be suggested that one of the chloropropenyl fragments in bis(chloropropenyl)sulfide **1** is dehydrochlorinated to afford allenic or methylacetylenic intermediates, while the second chloropropenyl moiety participates in the formation of a thiiranium intermediate **C**.

Further, intramolecular cyclization occurs via the interaction of electrophilic thiiranium fragment with relatively nucleophilic unsaturated counterpart. It cannot be ruled out that under the reaction conditions nucleophilic assistance of a hydrazine molecule takes place. The latter can probably attack the acetylene group, thus promoting further cyclization (intermediates **D**, **E**, **F**) into corresponding hydrazone **5**.

1,2-Migration of sulfur atom in this reaction (Figure 3) differs from that of migration of sulfur described in the literature. The fundamental difference lies in the fact that in case of the known literature data (see for example, [25]), the substituent at the sulfur atom is “passive”, i.e., it migrates together with the sulfur atom but is not directly involved in the chemical transformation.

In contradistinction to the literature data [1,2,3,4,5,6,7,8,9,10,11,12,13,14,15,16,17,18,19,20,21,22,23,24,25,26,27,28,29], cyclic product **5** (Figure 3) is formed due to the process when the substituent at the sulfur atom is not simply transferred with the chalcogen, but undergoes a series of cascade transformations.

Additionally, to our knowledge, the literature lacks the data on the 1,2-migration of sulfur “along” unsaturated group with the participation of the Cl-Csp^2^ fragment. The classical variant is a formation of thiiranium intermediate via intramolecular nucleophilic substitution of a halogen at tetragonal carbon atom with sulfur. Apart from the sulfur-containing acetylenic ketones and aldehydes (or their imines), which are transformed in situ into allenic derivatives followed by migration of sulfur in the presence of copper catalysts upon heating [25]. In the process presented herein (Figure 3), the formation of heterocyclic product **5** occurs as a result of 1,2-migration of sulfur “along” the chloropropenyl fragment with the participation of the Csp^2^ carbon atom under relatively mild conditions and without a metal-containing catalyst, which has not been previously noted in the literature.

The formation of pyrrole **6** can be tentatively rationalized as follows. Hydrazone **5** undergoes recyclization with ring-opening of compound **5** at the C–S bond to deliver a thiohydrazide fragment and an allene intermediate **G**, followed by another cyclization with participation of a nitrogen atom and further aromatization. Eventually, one of the key intermediates of this process, pyrrolothiol H, acting as an S-nucleophile, attacks bis(ethynyl)sulfide I (or possible tautomer), which, apparently, is formed under the reaction conditions from bis(chloropropenyl)sulfide **1**. In this case, the addition of S-nucleophile H to the sulfur-containing acetylene I proceeds according to the *trans*-addition rule to give the adduct **6** as *Z*-izomer (Figure 4).

The structure of compounds **5** and **6** has been proven by physical-chemical methods, including NMR spectroscopy and gas chromatography-mass spectrometry. The composition is confirmed by elemental analysis and molecular ion values.

So, the ^1^H NMR spectra of compound **5** contain multiplets of protons of the methyl, methylene, and methine groups of the heterocyclic ring in a higher field (1.18–2.95 ppm). The doubled broadened singlet of the NH_2_ group is observed at 4.8 ppm, in a lower field at 5.09 and 5.21 ppm, and separate signals of the protons of the terminal vinyl group are detected. In the ^13^C NMR spectra of compound **5**, signals of the methyl, methine, and methylene group, carbons appear at 17.6, 41.5, and 43.0 ppm, respectively. Signals of the vinyl group carbon and hydrazone fragment are observed in a weaker field. Absence of cross-peaks in the NOESY spectra between protons of the amino group and the methyl group allows to conclude that compound **5** in solutions exists as the *Z*-isomer relative to the C=N bond.

The ^1^H NMR spectra of heteroatom-containing polyunsaturated compound **6** show four signals of the methyl group protons, two signals of the methine protons, and a broadened singlet of the NH_2_ protons. In the ^13^C NMR spectra, signals of the methyl, propynyl, propenyl, and pyrrole carbons are detected. The presence of a cross-peak between the protons of the methyl and methine groups of the propenyl fragment in the NOESY spectra confirms the *cis*-position of sulfur atoms, i.e., the *Z*-configuration of the substituted vinyl group in compound **6**.

To additionally prove the unexpected thiophane structure of compound **5**, a series of new azine derivatives **7a**–**g** were synthesized (Figure 5).

According to the data of NMR spectroscopy, azines **7** in solutions exist as one isomer. Judging from XRD analysis of compound **7d**, azines **7** are apparently formed as isomers in which S-C=N-N fragment has *Z*-configuration and N-N=C-R^2^ fragment has *E*-configuration. X-Ray diffraction analysis of a crystal of azine **7d** (Figure 1) unambiguously confirms the thiophane structure of compounds **5** and **7**.

The azine fragment C2-N8-N9-C10 is planar and is characterized by the following parameters: bond lengths are C2=N8 1.274 (3) Å, C10=N9 1.258 (3) Å and N8-N9 1.414 (3) Å, torsion angle C2-N8-N9-C10 174.2 (2)°. The chlorophenyl substituent and the thiophane cycle lie in the plane of the azine fragment C2-N8-N9-C10; however, the S1 and C4 atoms of the thiophane cycle are out-of-plane by 0.105 Å and 0.600 Å, respectively. The thiophane cycle takes the envelope conformation. The nitrogen atoms of the azine fragment C2-N8-N9-C10 form two intramolecular hydrogen bonds C16-H16⋯N9 and C7-H7⋯N9 with the proton H16 of the aromatic ring and with one of the protons (H7) of the methyl group, respectively (Figure 1). It is important to note that in azine **7d**, due to such intramolecular hydrogen bonds, the configuration of the *1E,2Z* isomers is stabilized.

It should also be noted that an interesting point of the studied azine **7d** is the formation of dimeric structures under the condition of the implementation of intermolecular pnictogen binding of two adjacent **7d** molecules in the solid state (Figure 2). This type of interaction is possible due to the presence of short contacts N9 ⋯ Cg (3.677 Å) between the nitrogen atom of the azine fragment of one **7d** molecule and the phenyl substituent of the other.

It should be noted that azines find wide application as biologically active substances and reagents for organic synthesis [42,43,44,45]. Therefore, new compounds **7a**–**g** essentially expands the scope of the azine derivatives.

To synthesize new derivatives of the thiophene series, we have tried to remove the hydrazone fragment from compound **5** under the action of selenium dioxide. In this case, hydrazine is actually eliminated, but the heterocyclic fragment is further oxidized that ultimately furnishes 4-hydroxy-3-methyl-5-methylidenedihydrothiophen-2-one **8** (in acetonitrile) or 3-methyl-5- methylidenthiophen-2-one **9** (in dimethylformamide) (Figure 6).

Compounds **8** and **9,** which appear to be products of the Riley reaction, have been isolated in moderate yields.

Product **8**, due to the presence of two asymmetric carbon atoms, exists as diastereomers with *cis*- and *trans*-position of the hydroxyl and methyl groups that is confirmed by two groups of signals in the ^1^H and ^13^C NMR spectra in a 1:1 ratio. Taking into account the correlation between constant of proton spin-spin coupling and dihedral angle (Karplus equation), one can expect than the value of *^3^J_H3-H4_* for the *cis*-isomers of **8** would be higher than for the *trans*-isomers. This allows signals of the isomers in the NMR spectra to be assigned.

To further confirm the structure of 1-aminopyrrole **6**, we have implemented its reaction with 4-chlorobenzaldehyde, which affords the condensation product, the corresponding methylideneaminopyrrole **10** (Figure 7).

The structure of compound **10** was unambiguously proved by ^1^H and ^13^C NMR technique.

## 3. Materials and Methods

### 3.1. General Information

All solvents, hydrazine hydrate, KOH were purchased from commercial sources. Hydrazine hydrate 64% (monohydrate) technical grade was used. Bis(2-chloroprop-2-en-1-yl)sulfide was obtained according to the procedure [30,31]. The ^1^H, ^13^C, and ^15^N NMR spectra were recorded in CDCl_3_ solutions at room temperature on Bruker DPX-400 and AV-400 spectrometers (400.13, 100.61 and 40.56 MHz, respectively). ^1^H, ^13^C, and ^15^N chemical shifts (δ in ppm) were measured with accuracy of 0.01, 0.02, and 0.1 ppm, respectively. The residual solvent peak, δ_H_ = 7.27 and δ_C_ = 77.16 for CDCl_3_, δ_H_ = 2.50 and δ_C_ = 39.52 for DMSO-*d_6_*, and signal of nitromethane (^15^N) were used as references. Coupling constants (*J*) are reported in Hertz (Hz). The multiplicity abbreviations used are: s, singlet; d, doublet; t, triplet; q, quartet; m, multiplet; br, broad signal. Chromato-mass spectrometry analysis was performed on a Shimadzu GCMS-QP5050A mass spectrometer (EI ionization, 70 eV). The IR spectra of the compounds were recorded on a Varian 3100 FT-IR spectrometer with the sample in thin film or in KBr. Elemental analysis was performed on a Thermo Finnigan Flash series 1112 Elemental analyzer. Column chromatography was carried out on silica gel 60 (70–200 mesh; Merk).

### 3.2. Procedures for the Synthesis and Characterization Data for 4,5,9,10-Tetrahydrothieno [3’,4;5,6]cycloocta [1,2-c]thiophene (***2***), (Z)-1-(3-Methyl-5-Methylenedihydrothiophene-2(3H)-ylidene)hydrazine (***5***), and (Z)-3,5-Dimethyl-2-(1-(prop-1-ynylthio)prop-1-ene-2-ylthio)-1H-pyrrole-1-amine (***6***)

Bis(2-chloroprop-2-en-1-yl)sulfide (**1**) (4.60 g, 25 mmol) was added dropwise to a solution of KOH (5.63 g, 100 mmol) in hydrazine hydrate (15 mL) upon heating (40 °C) in 100 mL conical flask equipped with a reflux condenser with heated on magnetic stirrer. The reaction mixture was stirred at 50 °C for 3 h. Then water (50 mL) was added and the products were extracted with CHCl_3_ (3 × 30 mL). The extract was dried over MgSO_4_ and then evaporated. The products **2**, **5**, **6** were separated from the residue by column chromatography (eluent CHCl_3_).

*4,5,9,10-Tetrahydrothieno [3’,4;5,6]cycloocta [1,2-c]thiophene (**2**).* Yield 0.39 g (14%), mp 138–140 °C (mp 138–140 °C [32]). IR spectrum (KBr), ν, cм^−1^: 1439, 2863, 2923. ^1^H NMR, *δ*, ppm (CDCl_3_): 2.99 s (8H, 4CH_2_), 6.86 s (4H, H_thiophene_). ^13^C NMR, δ, ppm (CDCl_3_): 29.92 (CH_2_), 121.94 (C_thiophene_ -1,3,6,8), 142.15 (C_thiophene_ -3a,5a,8a,10a). MS, m/z: 220 [*M*]^+^. Found, %: C 65.83; H 5.40; S 28.75. C_12_H_12_S_2_. Calculated, %: C 65.41; H 5.49; S 29.10.

*(1E)-(3-Methyl-5-methylidenedihydrothiophen-2(3H)-ylidene)hydrazine (**5**).* Yield 2.41 g (68%), yellow oil. IR (film), ν, cm^−1^: 1617 (C=C, C=N), 3354 (NH_2_). ^1^H NMR, *δ*, ppm (CDCl_3_): 1.18 d (3H, *J* 6.6 Hz, CH_3_), 2.34 m and 2.83 m (2H, CH_2_), 2.91 m (1H, CH), 4.80 bs (2H, NH_2_), 5.09 s and 5.21 s (2H, =CH_2_). ^13^C NMR, *δ*, ppm (CDCl_3_): 17.61 (CH_3_), 41.49 (CH), 43.00 (CH_2_), 107.52 (=CH_2_), 144.01 (C=CH_2_), 157.03 (C=N). MS, *m/z*: 142 [*M*]^+^. Found, %: C 50.70; H 6.98; N 19.90; S 22.41. C_6_H_10_N_2_S. Calcd., %: C 50.67; H 7.09; N 19.70; S 22.55.

*3,5-Dimethyl-2-{[(Z)-1-methyl-2-(prop-1-yn-1-ylsulfanyl)ethenyl]sulfanyl}-1H-pyrrol-1-amine (**6**).* Yield 0.32 g (10%), yellow oil. IR (film), ν, cm^−1^: 1635 (C=C), 1696 (NH_2_), 2375 (C≡C), 3334 (NH). ^1^H NMR, *δ*, ppm (CDCl_3_): 1.73 d (3H, ^4^*J* 1.2 Hz, CH_3_CS), 2.00 s (3H, CH_3_C≡), 2.13 s (3H, 3-CH_3_), 2.27 s (3H, 5-CH_3_), 4.43 bs (2H, NH_2_), 5.73 s (1H, CH), 6.05 q (1H, ^4^*J* 1.2 Hz, SCH=). ^13^C NMR, *δ*, ppm (CDCl_3_): 5.01 (CH_3_C≡), 12.08 (CH_3_C-5), 12.42 (CH_3_C-3), 22.51 (CH_3_C=), 65.92 (CH_3_C≡C), 90.62 (CH_3_C≡), 105.40 (CH-4), 112.03 (CH_3_C-3), 119.49 (CH=), 125.76 (SC-2), 133.27 (CH_3_C-5), 134.39 (CH_3_C=). ^15^N NMR, *δ*, ppm (CDCl_3_): 208.3 (N-NH_2_), 317.3 (NH_2_). MS, *m/z*: 252 [*M*]^+^. Found, %: C 57.23; H 6.48; N 10.92; S 25.35. C_12_H_16_N_2_S_2_. Calcd., %: C 57.10; H 6.39; N 11.10; S 25.41.

### 3.3. General Procedure for the Synthesis and Characterization Data for Azines ***7***

A carbonyl compound (3.2 mmol) was added to compound **5** (0.45 g, 3.2 mmol) dissolved in MeOH (10 mL) in 100 mL conical flask equipped with a reflux condenser with heated on magnetic stirrer. The reaction mixture was boiled for 6 h. Then the solvent was evaporated. The products **7** were isolated by column chromatography (eluent CHCl_3_).

*(2Z)-1-(1-Methylethylidene)-2-(3-methyl-5-methylidenedihydrothiophen-2(3H)-ylidene)hydrazine (**7a**)*. From 0.19 g (3.2 mmol) of acetone. Yield 0.52 g (90%), colorless liquid. IR (film), ν, cm^−1^: 1599, 1621, 1643 (C=C, C=N). ^1^H NMR, *δ*, ppm (CDCl_3_): 1.30 d (3H, ^3^*J* 6.8 Hz, CH_3_CH), 1.99 s (3H, CH_3_), 2.04 s (3H, CH_3_), 2.43 m and 2.93 m (2H, CH_2_), 3.06 m (1H, CH), 5.08 s and 5.19 s (2H, =CH_2_). ^13^C NMR, *δ*, ppm (CDCl_3_): 18.13 (*Z*-CH_3_), 18.54 (CH_3_CH), 24.94 (*E*-CH_3_), 41.30 (CH), 42.69 (CH_2_), 106.74 (CH_2_=), 145.16 (C=CH_2_), 164.99 (C=N-1), 172.72 (C=N-2). MS, *m/z*: 182 [*M*]^+^. Found, %: C 59.43; H 7.91; N 15.50; S 17.15. C_9_H_14_N_2_S. Calcd., %: C 59.30; H 7.74; N 15.37; S 17.59.

*(1Z,2E)-1-(3-Methyl-5-methylidenedihydrothiophen-2(3H)-ylidene)-2-(1-phenylethylidene)hydrazine (**7b**).* From 0.38 g (3.2 mmol) of acetophenone. Yield 0.60 g (77%), white crystals, mp 49–50 °C. IR (film), ν, cm^−1^: 1610 (C=C, C=N). ^1^H NMR, *δ*, ppm (CDCl_3_): 1.40 d (3H, ^3^*J* 6.8 Hz, CH_3_CH), 2.44 s (3H, CH_3_), 2.51 m and 3.01 m (2H, CH_2_), 3.17 m (1H, CH), 5.12 s and 5.24 s (2H, =CH_2_), 7.41 m and 7.91 m (5H, Ph). ^13^C NMR, *δ*, ppm (CDCl_3_): 14.90 (CH_3_), 18.04 (CH_3_CH), 41.36 (CH), 42.67 (CH_2_), 106.73 (CH_2_=), 126.71 (C_o_), 128.15 (C_m_), 129.84 (C_p_), 137.81 (C_i_), 145.10 (C=CH_2_), 161.92 (C=N-2), 174.98 (C=N-1). MS, *m/z*: 244 [*M*]^+^. Found, %: C 68.53; H 6.68; N 11.52; S 13.25. C_14_H_16_N_2_S. Calcd., %: C 68.81; H 6.60; N 11.46; S 13.12.

*(1Z,2E)-1-(3-Methyl-5-methylenedihydrothiophen-2(3H)-ylidene)-2-(2-methylpropylidene)hydrazine (**7c**).* From 0.23 g (3.2 mmol) of 2-methylpropanal. Yield 0.44 g (70%), colorless liquid. IR (film), ν, cm^−1^: 1593, 1629 (C=C, C=N), 2874–2965 (C-H_Alk_). ^1^H NMR, *δ*, ppm (CDCl_3_): 1.14 d (6H, ^3^*J* 6.9 Hz, 2CH_3_), 1.31 d (3H, ^3^*J* 6.7 Hz, CH_3_), 2.46 m (1H, CH_2_), 2.61 m [1H, CH(CH_3_)_2_], 2.95 m (1H, CH_2_), 3.07 m (1H, CHCH_3_), 5.10 s and 5.21 s (2H, =CH_2_), 7.67 d (1H, ^3^*J* 5.4 Hz, CH=N). ^13^C NMR, *δ*, ppm (CDCl_3_): 18.03 (CH_3_), 19.58 (CH_3_), 31.83 (CH), 41.35 (CH), 42.76 (CH_2_), 106.98 (=CH_2_), 145.02 (C=CH_2_), 167.99 (N=CH), 177.10 (C=N). MS, *m/z*: 196 [*M*]^+^. Found, %: C 61.31; H 7.98; N 14.22; S 16.46. C_10_H_16_N_2_S. Calcd., %: C 61.18; H 8.21; N 14.27; S 16.33.

*(1E,2Z)-1-(4-Chlorobenzylidene)-2-(3-methyl-5-methylenedihydrothiophen-2(3H)-ylidene)-hydrazine (**7d**).* From 0.45 g (3.2 mmol) of 4-chlorobenzaldehyde. Yield 0.69 g (81%), yellow powder, mp 57–59 °C. IR (KBr), ν, cm^−1^: 1584, 1617 (C=C, C=N). ^1^H NMR, *δ*, ppm (CDCl_3_): 1.34 d (3H, ^3^*J* 6.7 Hz, CH_3_), 2.50 m and 2.98 m (2H, CH_2_), 3.14 m (1H, CHCH_3_), 5.12 s and 5.24 s (2H, =CH_2_), 7.37, 7.71 m (4H, C_6_H_4_), 8.35 s (1H, N=CH). ^13^C NMR, *δ*, ppm (CDCl_3_): 18.10 (CH_3_), 41.53 (CHCH_3_), 42.71 (CH_2_), 107.26 (=CH_2_), 129.00 (C_o_), 129.54 (C_m_), 132.56 (CCl), 137.01 (C_i_), 144.77 (C=CH_2_), 157.69 (N=CH), 179.98 (C=N). MS, *m/z*: 266 [*M*]^+^. Found, %: C 59.05; H 4.82; Cl 13.54; N 10.52; S 12.03. C_13_H_13_ClN_2_S. Calcd., %: C 58.97; H 4.95; Cl 13.39; N 10.58; S 12.11.

*(1E,2Z)-1-(4-Nitrobenzylidene)-2-(3-methyl-5-methylenedihydrothiophen-2(3H)-ylidene)hydrazine (**7e**).* From 0.48 g (3.2 mmol) of 4-nitrobenzaldehyde. Yield 0.70 g (80%), yellow powder, mp 134–136 °C. IR (KBr), ν, cm^−1^: 1340, 1513 (NO_2_), 1552, 1599 (C=C, C=N). ^1^H NMR, *δ*, ppm (DMSO-d_6_): 1.28 d (3H, ^3^*J* 6.6 Hz, CH_3_), 2.53 m and 3.04 m (2H, CH_2_), 3.20 m (1H, CHCH_3_), 5.19 s and 5.35 s (2H, =CH_2_), 8.05 and 8.33 m (4H, C_6_H_4_), 8.64 s (1H, N=CH). ^13^C NMR, *δ*, ppm (DMSO-d_6_): 17.59 (CH_3_), 41.09 (CHCH_3_), 41.94 (CH_2_), 107.80 (=CH_2_), 124.06 (C_m_), 129.12 (C_o_), 139.63 (C_i_), 144.23 (C=CH_2_), 148.72 (CNO_2_), 157.26 (N=CH), 180.58 (C=N). MS, *m/z*: 275 [*M*]^+^. Found, %: C 56.85; H 4.65; N 15.32; S 11.53. C_13_H_13_N_3_O_2_S. Calcd., %: C 56.71; H 4.76; N 15.26; S 11.65.

*(1E,2Z)-1-(2-Nitrobenzylidene)-2-(3-methyl-5-methylenedihydrothiophen-2(3H)-ylidene)hydrazine (**7f**).* From 0.48 g (3.2 mmol) of 2-nitrobenzaldehyde. Yield 0.63 g (72%), yellow powder, mp 66–68 °C. IR (KBr), ν, cm^−1^: 1343, 1518 (NO_2_), 1574, 1609 (C=C, C=N). ^1^H NMR, *δ*, ppm (CDCl_3_): 1.35 d (3H, ^3^*J* 6.7 Hz, Me), 2.51 m and 3.00 m (2H, CH_2_), 3.16 m (1H, CHCH_3_), 5.11 s and 5.25 s (2H, =CH_2_), 7.55, 7.65, 7.99, and 8.17 m (4H, C_6_H_4_), 8.86 s (1H, N=CH). ^13^C NMR, *δ*, ppm (CDCl_3_): 18.12 (CH_3_), 41.81 (CHCH_3_), 42.73 (CH_2_), 107.55 (=CH_2_), 124.66 (C_m_), 128.97 (C_i_), 129.60 (C_o_), 131.09 (C_p_), 133.37 (C_m_), 144.67 (C=CH_2_), 148.93 (CNO_2_), 154.61 (N=CH), 181.50 (C=N). MS, *m/z*: 275 [*M*]^+^. Found, %: C 56.91; H 4.70; N 15.30; S 11.47. C_13_H_13_N_3_O_2_S. Calcd., %: C 56.71; H 4.76; N 15.26; S 11.65.

*(1E,2Z)-1-(2,4-Dimethoxybenzylidene)-2-(3-methyl-5-methylenedihydrothiophen-2(3H)-ylidene)hydrazine (**7g**).* From 0.53 g (3.2 mmol) of 2,4-dimethoxybenzaldehyde. Yield 0.81 g (87%), yellow oil. IR (film), ν, cm^−1^: 1609, 1676 (C=C, C=N). ^1^H NMR, *δ*, ppm (CDCl_3_): 1.36 d (3H, ^3^*J* 6.9 Hz, CH_3_), 2.48 m and 2.98 m (2H, CH_2_), 3.14 m (1H, CHCH_3_), 3.82 s (3H, CH_3_O), 3.83 s (3H, CH_3_O), 5.12 s and 5.23 s (2H, =CH_2_), 6.42, 6.53, 8.01 m (3H, C_6_H_3_), 8.77 s (1H, N=CH). ^13^C NMR, *δ*, ppm (CDCl_3_): 18.09 (CH_3_), 41.19 (CH_2_), 42.72 (CHCH_3_), 55.35 (OCH_3_), 55.49 (OCH_3_), 97.92 (C_m_), 105.68 (C_m’_), 106.67 (=CH_2_), 115.66 (C_i_), 128.58 (CH_o_), 145.19 (C=CH_2_), 154.72 (N=CH), 160.33 (C_o’_), 163.46 (C_p_), 177.02 (C=N). MS, *m/z*: 290 [*M*]^+^. Found, %: C 61.97; H 6.37; N 9.50; S 11.20. C_15_H_18_N_2_O_2_S. Calcd., %: C 62.04; H 6.25; N 9.65; S 11.04.

### 3.4. Procedure for the Synthesis and Characterization Data for 4-Hydroxy-3-methyl-5-methylenedihydrothiophen-2(3H)-one (***8***)

Compound **5** (0.50 g, 3.4 mmol) was dissolved in MeCN (20 mL), then SeO_2_ (0.38 g, 3.4 mmol) was added. The reaction mixture was stirred for 10 h. Then, the solvent was evaporated. The product **8** was isolated by column chromatography (eluent Et_2_O: hexane = 2:1) as two diastereomers.

Yield 0.30 g (57%), yellow liquid. IR (film), ν, cm^−1^: 1628 (C=C), 1717 (C=O), 3413 br (OH). ^1^H NMR, *δ*, ppm (CDCl_3_): 1.26 d (3H, ^3^*J* 6.0 Hz, CH_3_), 1.27 d (3H, ^3^*J* 6.0 Hz, CH_3_), 2.68 m (1H, CHCH_3_), 2.81 m (1H, CHCH_3_), 4.43 d (1H, ^3^*J* 7.6 Hz, CHOH), 4.77 d (1H, ^3^*J* 6.4 Hz, CHOH), 5.34 s and 5.36 s (2H, CH_2_=), 5.66 s and 5.67 s (2H, CH_2_=). ^13^C NMR, *δ*, ppm (CDCl_3_): 9.07, 13.43 (Me), 53.11, 54.59 (CHCH_3_), 75.74, 78.96 (CHOH), 109.80, 111.67 (CH_2_=), 144.96, 145.69 (C=CH_2_), 203.40, 205.34 (C=O). MS, *m/z*: 144 [*M*]^+^. Found, %: C 50.07; H 5.77; S 22.10. C_6_H_8_O_2_S. Calcd., %: C 49.98; H 5.59; S 22.24.

### 3.5. Procedure for the Synthesis and Characterization Data for 3-Methyl-5-methylenethiophen-2(5H)-one (***9***)

To compound **5** (0.90 g, 6.3 mmol), dissolved in DMF (30 mL), SeO_2_ (0.7 g, 63 mmol) was added. The reaction mixture was stirred for 3 h at 100 °C. Then, the reaction mixture was cooled, and the precipitate was separated by filtration. The filtrate was mixed with water (50 mL). The raw product was extracted with CHCl_3_ (3 × 20 mL). The extract was evaporated. The product **9** was isolated by column chromatography (eluent Et_2_O: hexane = 1:4).

Yield 0.24 g (30%), yellow oil. IR (film), ν, cm^−1^: 1623 (C=C), 1681 (C=O). ^1^H NMR, *δ*, ppm (CDCl_3_): 1.97 s (3H, CH_3_), 5.59 s and 5.77 s (2H, =CH_2_), 7.23 s (1H, CH=). ^13^C NMR, *δ*, ppm (CDCl_3_): 11.89 (CH_3_), 116.59 (CH_2_=), 142.26 (C=CH_2_), 142.37 (CCH_3_), 144.09 (CH), 196.38 (C=O). MS, *m/z*: 126 [*M*]^+^. Found, %: C 57.03; H 4.77; S 25.20. C_6_H_6_OS. Calcd., %: C 57.11; H 4.79; S 25.41.

### 3.6. Procedure for the Synthesis and Characterization Data for N-[(4-Chlorophenyl)methylidene]-3,5-dimethyl-2-{[(Z)-1-methyl-2-(prop-1-yn-1-ylsulfanyl)ethenyl]sulfanyl}-1H-pyrrol-1-amine (***10***)

To compound **6** (1.10 g, 4.4 mmol), dissolved in MeOH (10 mL), 4-chlorobenzaldehyde (0.62 g, 4.4 mmol) was added. The reaction mixture was refluxed for 8 h. Then, the solvent was evaporated under reduced pressure. The residue was purified by column chromatography (eluent CHCl_3_).

Yield 0.24 g (67%), yellow oil. ^1^H NMR, *δ*, ppm (CDCl_3_): 1.69 s (3H, CH_3_C≡), 1.97 d (3H, ^4^*J* 1.8 Hz, CH_3_CS), 2.19 s (3H, CH_3_-3), 2.34 s (3H, CH_3_-5), 5.90 s (1H, CH-4), 6.01 d (1H, ^4^*J* 1.8 Hz, =CH), 7.42, 7.82 m (4H, C_6_H_4_), 8.74 s (1H, CH=N). ^13^C NMR, *δ*, ppm (CDCl_3_): 5.01 (CH_3_C≡), 12.82 (CH_3_-5), 12.88 (CH_3_-3), 22.34 (CH_3_C=), 65.90 (CH_3_C≡C), 90.49 (CH_3_C≡), 107.59 (CH-4), 108.11 (CH_3_C-3), 118.07 (CH=), 128.99 (C_m_), 129.36 (C_o_), 130.09 (SC-2), 132.84 (CCl), 133.87 (C_i_), 135.79 (CH_3_C-5), 136.72 (CH_3_C=), 154.62 (C=N). Found, %: C 61.02; H 4.77; Cl 10.02; N 5.01; S 16.53. C_19_H_19_ClN_2_S_2_. Calcd., %: C 60.86; H 5.11; Cl 9.46; N 5.47; S 17.10.

### 3.7. X-ray Crystallography

X-ray data were collected on a BRUKER D8 VENTURE PHOTON 100 CMOS diffractometer with MoK_α_ radiation (λ = 0.71073 Å) using the φ and ω scans technique. Using Olex2 [46], the structure was solved with the ShelXS [47] structure solution program using Direct Methods and refined with the XL [47] refinement package using least squares minimization. Data were corrected for absorption effects using the multi-scan method (SADABS) [48]. All non-hydrogen atoms were refined anisotropically using SHELX [47]. The coordinates of the hydrogen atoms were calculated from geometrical positions. Crystallographic experimental details, bond lengths, bond angles and torsion angles for compound **7d** are given in Appendix A. These data can be obtained free of charge from The Cambridge Crystallographic Data Centre via http://www.ccdc.cam.ac.uk access date 25 October 2021 (CCDC number 2117593).

## 4. Conclusions

In conclusion, a new heterocyclization of bis(2-chloroprop-1-en-3-yl)sulfide in hydrazine hydrate–alkali has been discovered. The reaction probably involves the unexpected 1,2-migration of the sulfur atom along the chloroalkenyl fragment followed by heterocyclization and other processes leading to thiophen-2-one and 1-aminopyrrole derivatives. These reactions open up expedient approaches to the synthesis of various hardly accessible thiophene and pyrrole compounds from 2,3-dichloropropene and elemental sulfur as starting reagents.

## Data Availability

The data presented in this study are available in the Appendix A.

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
