# Peer review of "Heterocyclization of Bis(2-chloroprop-2-en-1-yl)sulfide in Hydrazine Hydrate–KOH: Synthesis of Thiophene and Pyrrole Derivatives"

_molecules, 2022, doi:10.3390/molecules27206785_

Round 1
Reviewer 1 Report
This work reported a unique transformation of bis(2-chloroprop-2-en-1yl)sulfide 1 in the presence of hydrazine and KOH, affording the dihydrothiophen derivative 5. More data would be necessary to show synthetic utility and/or mechanistic novelty of this reaction, because only net one reaction example was demonstrated.In addition, the unique mechanism involving the thiiranium intermediates shown in Scheme 3 should be important to explain the novelty of this work as explained also in the introduction section by the authors. However, examination is insufficient to prove this mechanism: e.g. the three-membered ring intermediate can really form? What about the reaction of the intermediate A with hydrazine, followed by cyclization and ring-rearrangement? According to the proposed mechanism, other nucleophiles instead of hydrazine can be used, which will greatly improve this work from the viewpoint of not only the synthetic utility but also generality of this transformation.
Reviewer 2 Report
The Rozentsveig lab reports an interesting intramolecular cyclization of bis(2-chloroprop-2-en-1-yl)sulfide through the action of hydrazine hydrate/KOH system. Along with expected dithiophenocyclooctane, the authors isolated a previously unknown hydrazone of 5-methylidene-3-methyl-dihydrothiophen-2-one, whose formation is very interesting from a mechanistic point of view. The reaction presumably proceeds through elimination of two equivalents of HCl to form thiiranium intermediate which undergoes a 1,2-migration of sulfur atom to a five-membered cyclic sulfur-containing intermediate readily converted to the final product by prototropic isomerization. The proposed mechanism is reasonable enough, although I have doubts about the intermediate with three separated charges in Scheme 3. The authors should clarify this point. The products were mainly characterized by all required methods, however compounds 8 and 10 have impurities and should be repurified. Also, the type of hydrazine hydrate solution and equipment for the main reaction should be added. I thus recommend publication in Molecules after addressing above comments.
Round 2
Reviewer 1 Report
The response comments given by the authors contained important information to emphasize the uniqueness and specificity of the presented transformation. I think after briefly describing that discussion and information including the negative results in the manuscript, it can meet criteria for publication for Molecules.